# Isolation and Identification of *Microvirga thermotolerans* HR1, a Novel Thermo-Tolerant Bacterium, and Comparative Genomics among *Microvirga* Species

**DOI:** 10.3390/microorganisms8010101

**Published:** 2020-01-10

**Authors:** Jiang Li, Ruyu Gao, Yun Chen, Dong Xue, Jiahui Han, Jin Wang, Qilin Dai, Min Lin, Xiubin Ke, Wei Zhang

**Affiliations:** 1School of Life Science and Engineering, Southwest University of Science and Technology, Mianyang 621010, Sichuan, China; lijiangemail01@126.com (J.L.); wangjin@caas.cn (J.W.); daiqilinmj@sina.com (Q.D.); 2Biotechnology Research Institute, Chinese Academy of Agricultural Sciences, Beijing 100081, China; 82101182011@caas.cn (R.G.); chenyun0402ye@163.com (Y.C.); xue_dong_kevin@126.com (D.X.); 13121257599@163.com (J.H.); linmin57@vip.163.com (M.L.)

**Keywords:** isolation, identification, *Microvirga thermotolerans* HR1, comparative genomics

## Abstract

Members of the *Microvirga* genus are metabolically versatile and widely distributed in Nature. However, knowledge of the bacteria that belong to this genus is currently limited to biochemical characteristics. Herein, a novel thermo-tolerant bacterium named *Microvirga thermotolerans* HR1 was isolated and identified. Based on the 16S rRNA gene sequence analysis, the strain HR1 belonged to the genus *Microvirga* and was highly similar to *Microvirga* sp. 17 mud 1-3. The strain could grow at temperatures ranging from 15 to 50 °C with a growth optimum at 40 °C. It exhibited tolerance to pH range of 6.0–8.0 and salt concentrations up to 0.5% (*w/v*). It contained ubiquinone 10 as the predominant quinone and added group **8** as the main fatty acids. Analysis of 11 whole genomes of *Microvirga* species revealed that *Microvirga* segregated into two main distinct clades (soil and root nodule) as affected by the isolation source. Members of the soil clade had a high ratio of heat- or radiation-resistant genes, whereas members of the root nodule clade were characterized by a significantly higher abundance of genes involved in symbiotic nitrogen fixation or nodule formation. The taxonomic clustering of *Microvirga* strains indicated strong functional differentiation and niche-specific adaption.

## 1. Introduction

*Microvirga subterranea* is a typical species, so Kanso and Patel initially established the genus *Microvirga* (family Methylobacteriaceae, order Rhizobiales, class Alphaproteobacteria) [1,2]. After that, the description of the genus *Microvirga* was amended three times. Zhang and Song expanded the optimum temperature range and temperature range for growth on the basis of Kanso and Patel [1,2], Hang-Yeon and Soon-Wo added the G+C content of the DNA, the predominant isoprenoid quinone, and the major fatty acid information into the description of the genus *Microvirga* [3], and the latest revision was made by Zhang and Zhang, stating that the nitrate reduction was variable, C_18_: _1_ω7c and/or C_19: 0_ cyclo ω8c was contained in the major fatty acids, and the genome size and DNA G+C content were 3.53–9.63 Mb and 61.1–65.1%, respectively [4]. Generally the members of *Microvirga* showed common features in the appearance of cells and the composition of their cell wall. Sixteen strains have been isolated from various environmental niches and have been included in the genus *Microvirga*, including *Microvirga aerilat**a* 5420S-16^T^ and *Microvirga aerophila* 5420S-12^T^, which were isolated from air [3], *Microvirga arabica* SV2184P^T^, *Microvirga makkahensis* SV1470^T^, *Microvirga guangxiensis* CGMCC 1.7666^T^*, Microvirga indica* S-MI1b^T^*, Microvirga pakistanensis* NCCP-1258^T^*, Microvirga soli* R491^T^*, Microvirga rosea* MAH-2^T^*,* and *Microvirga flavescens* c27j1^T^*,* which were isolated from different soil types [1,4,5,6,7,8], *Microvirga flocculans* TFB^T^*,* which was isolated from a hot spring [3,9], *Microvirga lotononidis* WSM3557^T^*, Microvirga lupine* Lut6^T^*,* and *Microvirga zambiensis* WSM3693^T^*,* which were isolated from nitrogen-fixing nodules of the legumes [10], *Microvirga ossetica* V5/3M^T^ and *Microvirga vignae* BR3299^T^*,* which were isolated from root nodules [11,12], *Microvirga massiliensis* JC119^T^*,* which was isolated from a stool sample [13], and *Microvirga brassicacearum* CDVBN77^T^*,* which was isolated from rapeseed [14].

Previous studies have mainly focused on comparisons among the *Microvirga* species according to their physical characteristics and biological functions. For example, *M. aerilata, M. aerophila, M. guangxiensis, M. pakistanensis, M. soli, M. rosea, M. lupine,* and *M. massiliensis* have flagellum, but the others species do not. Moreover, *M. lupine*, *M. zambiensis,* and *M. vignae* could form unique colorful colonies that are distinct from other species [10]. In addition, *M. indica* S-MI1b could oxidize arsenite in a broad pH range from 4.0 to 9.0 [6]. Furthermore, *M. subterranean* FaiI4, an only moderate thermophile bacterium, could grow over a wide temperature range from 25 to 45 °C and grow optimally at 41 °C [2]. To date, there are 16 valid species in the genus *Microvirga* listed in the LPSN database (http://www.bacterio.net/), and the complete genome of *Microvirga* sp. 17 mud 1–3 has been published [15]. According to the analysis of this complete genome, for instance, *Microvirga* sp. 17 mud 1–3 is considered a radiation-resistant bacterium [15], *M. zambiensis*, *M. lupini*, *M. lotononidis*, *M. guangxiensis*, and *M. vignae* BR 3299^T^ are considered symbiotic nitrogen-fixing alphaproteobacteria [16], and *M. brassicacearum* CDVBN77 is considered a rapeseed endophyte with biotechnological potential [14].

In nature, *Microvirga* species have been found in a wide range of ecological niches. Studies of culturable bacteria belonging to the *Microvirga* genus typically focus on isolation, identification, and phenotypical characterization, while little attention has been paid to the genetic make-up of the *Microvirga* genus and to comparing their phenotypes. The present study describes the novel thermo-tolerant bacterium *Microvirga thermotolerans* HR1 isolated from a rice paddy located in the main rice-producing area, Wuchang, in the northeast of China. Furthermore, the whole genome sequencing of the *Microvirga thermotolerans* HR1 strain and comparative genomic analysis of 11 strains of *Microvirga* are provided to explain the specific niche adaptation of various *Microvirga* strains.

## 2. Materials and Methods

### 2.1. Sample Collection, Strain Isolation, and Cultivation

The rhizospheric soil sample was collected from Wuchang, the main rice-producing area in the northeast of China (N 45° 3′7″, E 127° 3′24″). To isolate the bacteria, 2 g of soil sample was added to 20 mL of R2A, LB, TGY liquid medium and incubated at 16, 30, 37, and 48 °C for 3 days. Then, 200 μL of enrichment solution was transferred into 20 mL of fresh R2A liquid medium correspondence with last step and incubated at 16, 30, 37, and 48 °C for 3 days. This routine was repeated three times to obtain an enrichment solution. Afterward, 100 μL of enrichment solution was diluted for 10^−3^, 10^−4^, and 10^−5^, respectively, with PBS buffer (KH_2_PO_4_ 0.2 g, Na_2_HPO_4_·12H_2_O 2.9 g, NaCl 8 g, KCl 0.2 g, pH = 7.0). The dilutions were plated on correspondence agar plates and incubated at 16, 30, 37, and 48 °C. After 3 days, the colony was selected for 16S rRNA sequencing. The strain HR1 was isolated from R2A agar plates incubated at 48 °C.

### 2.2. 16S rRNA and Housekeeping Gene Amplification and Analysis

Genomic DNA of strain HR1 was extracted and purified by a commercial bacterial genomic DNA isolation kit (Magen, Guangzhou, China). The 16S rRNA gene was amplified with the universal bacterial primers F27 and R1492 [17], and housekeeping genes (*gyrB*, *recA*, and *rpoB*) were amplified with the universal primers described by Radl and Simoes-Araujo [12]. The amplified fragments were cloned into a cloning-vector pJET1.2/Blunt Vector (Thermo Scientific, Waltham, MA, USA) and sequenced by BGI ARK Biotechnology (Beijing, China). A preliminary 16S rRNA gene sequence analysis was performed using the Ezbiocloud server (https://www.ezbiocloud.net/identify) [18], and further phylogenetic analyses were performed through the software MEGA 7.0 [19] with the neighbor-joining method [20]. Bootstrap values were calculated based on 1000 replicates. The housekeeping gene (*gyrB, recA,* and *rpoB*) phylogenetic analyses were also used to confirm the phylogenetic relationship of strain HR1 with the neighbor-joining [20]. Bootstrap values were calculated based on 1000 replicates. According to the 16S rRNA sequence analysis, the closely related type strains were bought from GDMCC and were used for chemotaxonomic analysis with HR1. The GenBank accession number of 16S rRNA sequence was MN524586.

### 2.3. Physiological Characteristics of Microvirga thermotolerans HR1

To investigate the optimum condition for HR1 growth, strain HR1 was cultivated on a series of different growth conditions. The growth test was performed on R2A agar (and liquid), LB agar (and liquid), TGY agar (and liquid), and Rouf’s liquid medium (1 g of yeast extract, 5 g of peptone, 0.2 g of MgSO_4_·7H_2_O, 0.05 g of CaCl_2_, 0.15 g of ferric ammonium citrate, 0.05 g of MnSO_4_·4H_2_O, 0.01 g of FeCl_3_·4H_2_O, 17 g of agar, 10 mL of vitamin solution, and 1 mL of trace-element solution) [2] at 4, 15, 20, 25, 30, 37, 40, 45, 50, and 55 °C for 3 days, respectively. A method for gram-stain reaction determination was modified from Buck’s method [21]. The HR1 strain was grown on R2A agar for 3 days at 40 °C for cell morphology and size observation using a transmission electron microscope (H-7650, Hitachi, Tokyo, Japan). The optimal temperatures for growth were investigated by growth on R2A agar at different temperatures (4–55 °C, interval as description above) for 7 days. Tolerance to different NaCl concentrations (0–0.5%, at intervals of 0.1%, *w/v*, NaCl) and pH range (pH 4.0–11.0, at intervals of 1 unit) were performed at 40 °C for 7 days. Anaerobic growth was tested in an MGCAnaeroPouch-Anaero (Mitsubishi, Tokyo, Japan) for 7 days at 40 °C on R2A agar. Catalase and oxidase activities were investigated in 3% (*v/v*) H_2_O_2_ and using commercial strips (Huankai, Guangzhou, China) according to the manufacturer’s instruction, respectively. Enzymatic and carbon source utilization assays were tested using the API 20NE, API ZYM (bioMerieux, Marcy-l’Etoile, French) and Biolog plates kits (Hayward, CA, USA) according to the manufacturers’ instruction after 7-day growth on R2A agar at 40 °C.

### 2.4. Chemotaxonomic Analysis of Microvirga thermotolerans HR1

To investigate the chemotaxonomic features of strain HR1, a series of experiments were carried out to determine the content of the respiratory quinones, polar lipids, and fatty acids of closely related or type strains (*M. flavescens* c27j1, *M. indica* S-MI1b and *M. subterranean* FaiI4) and HR1. Respiratory quinones of the studied strain were extracted and analyzed via the HPLC system. Polar lipids of strain HR1 were extracted and examined by two-dimensional TLC. The fatty acids were extracted, quantified, and analyzed using the microbial identification system with strain HR1 and related type strains grown on R2A agar for 3 days at optimal growth condition.

### 2.5. Complete Genome Sequencing and Analysis

After the cells were cultivated in Rouf’s medium at 40 °C overnight, a commercial bacterial genomic DNA isolation kit (Magen) was used to extract and purify the genome DNA of strain HR1. Concentration and quality of DNA was detected by Nanodrop2500 (OD_260_/ OD _280_ = 1.8–2.0, ≥10 μg).

The complete genome was sequenced using the Illumina Hiseq and PacBio platform. The assembly software Canu and SPAdes was used to assembly the complete genome of strain HR1. Glimmer (http://ccb.jhu.edu/software/glimmer/index.shtml) GeneMarkS and Prodigal software were used to predicted the coding sequence (CDS) in the genome of strain HR1. Gene functional annotation was mainly based on protein sequence alignment, and the corresponding functional annotation information was obtained by comparing the gene sequence with each database. Databases used include NR, swiss-prot, Pfam, EggNOG, GO, and KEGG. The ANI calculator (www.ezbiocloud.net/tools/ani) and the Genome-to-Genome Distance Calculator (GGDC 2.1) [22] (http://ggdc.dsmz.de/home.php) were used for calculated the average nucleotide identity [23] (ANI) and digital DNA–DNA hybridization (dDDH) values, respectively. The GenBank accession number of the complete genome was CP045423.

### 2.6. Comparative Genomics of Microvirga Species

The present study compared the genomes of 11 bacteria belonging to the genus *Microvirga*. The 11 bacteria and genome accession numbers were as follows: *M. massiliensis* JC119, LN811386; *M. vignae* BR3299, LCYG01000000; *M. subterranea* DSM 14364, QQBB01000000; *M. aerophila* NBRC 106136, BJYU00000000; *M. guangxiensis* CGMCC 1.7666, FMVJ01000000; *M. lotononidis* WSM3557, AJUA01000000; *M. brassicacearum* CDVBN77, VCMV01000000; *M. ossetica* V5/3M, CP016616; *Microvirga* sp. 17 mud 1–3, CP029481; *Microvirga* sp. KLBC 81, QDAH01000000; *M. thermotolerans* HR1, CP045423. The Bacterial Pan Genome Analysis (BPGA) pipeline [24] was used for the pan genome analyses. The clustering tool USEARCH was used to cluster protein families. A 50% sequence identity was considered as the cut-off value for orthologous clustering to obtain the pan and core genome. After obtaining the core genome of the *Microvirga* genus, the OrthoFinder [25,26] was used to perform an all-versus-all BLAST search and identify clusters of orthologous genes (OGs), and those OGs were then aligned and concatenated by MUSCLE [27]. A phylogenetic tree based on orthologous proteins of the *Microvirga* genus was constructed by FastTree [28] according to the maximum-likelihood method.

### 2.7. Effect of Temperature on the Growth of Microvirga thermotolerans HR1

*M. thermotolerans* HR1 and *M. subterranean* FaiI4 (as a control strain) were incubated to mid-exponential phase (OD_600_ = 0.2–0.4) in Rouf’s liquid medium [2]. Afterward, culture medium was added to 20 mL of fresh medium at a final OD_600_ value of 0.1, and incubated under different temperature conditions (15, 20, 25, 30, 37, 40, 45, and 50 °C at a pH set-point of 7.0). After 15 h incubation, bacterial growth was determined by using Spec.

## 3. Results

### 3.1. Isolation and Characterization of Microvirga thermotolerans HR1

The HR1 strain was isolated from paddy soil collected from the main rice-producing area in Wuchang, in the northeast of China. This isolate was routinely cultivated on R2A agar and in Rouf’s liquid medium at 40 °C for overnight. A milk white, semi-transparent, smooth, and drop-shaped substance appeared on the agar after 3 days of incubation. Staining experiments revealed that the isolate was Gram-negative. Microscopic examination confirmed that the cells were rod-shaped, and 0.7–0.9 µm wide and 1.2–2.8 µm long with flagellum (Figure 1a–c). Physiological analyses revealed that the isolate was able to grow at 15–50 °C and pH 6.0–8.0 and in the presence of 0–0.5% (*w/v*) NaCl, and optimal growth was achieved at 40 °C, pH 7.0, without NaCl. The rise of culture temperatures from 45 to 50 °C resulted in a dramatically decrease of the strain HR1. Similarly, a more extreme stress on *M. subterranean* FaiI4, which was previously described as thermophilic bacterium, was observed when the temperature reached up to 45–50 °C (Figure 2).

A positive reaction was observed for catalase but not for oxidase. Other phenotypic characteristics were detailed in the species description found in Table 1. 

The tests for nitrate reduction, hydrolysis of gelatin were negative; assimilation for arabinose, citric acid and phenylacetic acid were negative; the catalase was positive for alkaline phosphatase, esterase (C4), weakly positive for esterase lipase (C8), acid phosphatase, negative for lipid enzyme (C14). Carbon source utilization for tetrazolium violet was positive, minocycline was weakly positive, α-d-glucose, D-fucose, fusidic acid, myo-inositol, α-keto-glutaric acid, and tween 40 were negative. The differential characteristics of strain HR1 with respect to the most closely related species, *M. flavescens* c27j1, *M.indica* S-MI1b and the type strain of genus *M. subterranean* FaiI4 were also shown in Table 1.

The major respiratory quinone of strain HR1 was Q-10, which was the same as that of other species of the *Microvirga* genus. The polar lipids of strain HR1 included diphosphatidylglycerol (DPG), phosphatidylethanolamine (PE), phosphatidylglycerol (PG), phosphatidylcholine (PC), phospholipids (PL), and aminolipid (AL) (Appendix A). The constituents of quinone and polar lipids were consistent with the characteristics of the genus *Microvirga*. The major fatty acids (>10% of the total) of strain HR1 were C_18: 1_ ω7c and/or C_18: 1_ ω6c (56.76%) and C_19: 0_ cyclo ω8c (16.8%), compared with its closely related type strains. The comments of major fatty acids in strain HR1 was consistent with the type species (*M. subterranean* FaiI4), but different from other closely related type strains (Table 2).

### 3.2. Phylogenic Analysis of 16S rRNA and Housekeeping Genes

The analysis of 16S rRNA and housekeeping genes (*rpoB*, *recA*, and *gyrB*) was employed to elucidate the taxonomic position of strain HR1. The 16S rRNA gene (accession no: MN524586) and partly housekeeping gene (*gyrB*, *recA* and *rpoB*) sequence of strain HR1 comprised 1467, 882, 1016, and 965 bp, respectively. The 16S rRNA gene of strain HR1 shared the highest similarity to *Microvirga* sp. 17 mud 1–3 (97.87%), *M. flavescens* c27j1 (97.37%), and *M. aerilata* 5420S–16 (97.16%). The *rpoB*, *recA* and *gyrB* genes of strain HR1 shared the highest similarity (92.43, 92.59 and 89.88%, respectively) to *Microvirga* sp. 17 mud 1–3. Phylogenetic analysis based on the 16S rRNA gene constructed by the neighbor-joining method showed that strain HR1 was a member of genus *Microvirga* (Figure 3). Housekeeping genes phylogenetic analysis revealed that strain HR1 was clearly differentiated from other species of the genus *Microvirga* (Appendix A).

### 3.3. Genomic Features of Microvirga thermotolerans HR1 Strain

The complete genome sequence of strain HR1 was 3,823,049 bp, and DNA G+C content was 67.71%, containing 3818 putative protein-coding sequences (CDSs), 51 tRNA genes, and 6 rRNAs genes (Appendix A). This was similar to other *Microvirga* species (the genome size ranged from 3.53 to 9.63 Mbp, Table 3)*,* such as *Microvirga* sp. 17 mud 1–3, *M. flavescens* c27j1, *M. aerilata* 5420S-16, and *M. indica* S-MI1b. However, the strain showed high genome G+C content compared with the other *Microvirga* species (reference ranged from 61.1 to 65.1%) sequenced so far (Table 3). The digital DNA–DNA hybridization values of strain HR1 and *Microvirga* sp. 17 mud 1–3 and of that and *M. flavescens* c27j1 based on the whole genome sequence were 38.2 and 19.9%, while the ANI values were 84.21 and 77.67%, respectively. Therefore, complete genome analysis combined with 16S rRNA phylogenic, physiological, and biochemical properties all supported the identification of the strain HR1 as a novel species of the genus *Microvirga.*

### 3.4. Niche Adaption of Microvirga Species

A summary of 11 whole-genome comparisons between *M. thermotolerans* HR1 and the publicly available genome sequences of members of the genus *Microvirga* (downloaded from NCBI database, Table 3) were used for the comparative genomic analysis using the Bacterial Pan Genome Analysis (BPGA) pipeline. The size and G+C content of the genomes used in this study ranged from 3.8 to 9.1 MB and 61.1 to 67.71%, respectively. Generally, a local database containing 11 genomes and 57,302 putative protein-coding genes was created.

Based on this database, 1558 (2.71%) shared orthologous coding sequences were clustered into the core genome of *Microvirga*, 27,150 (47.38%) were represented in the accessory genome, and 12,549 (21.8%) were identified as strain-unique genes (Figure 4a). Therefore, a highly reliable mathematical extrapolation of the pan and core genome was constructed (Appendix A). The total genes increase in the pan genome of *Microvirga* with the rise in the analyzed genome number, suggesting that the pan genome was open. Meanwhile, the genes’ number of core genomes was highly conserved, relatively reaching a constant after five species were added to the analysis, indicating that the core genome of genus *Microvirga* was conserved.

Analysis of the distribution pattern of *Microvirga* strains based on 1558 core orthologous proteins generated two main distinct clusters: a mostly soil clade consisting of soil isolates and a predominantly root nodule clade consisting of nodule-formation bacteria or rhizobia (Figure 4b). In brief, three strains, i.e., *M. thermotolerans* HR1, *Microvirga* sp. 17 mud 1–3, and *M. subterranea* Fail4, formed a separate cluster, named a soil clade, and shared common features, e.g., a higher G+C content, a smaller genome size, thermo-tolerance, and radiation resistance. Notably, a markedly lower genome size and lower numbers of genes and proteins were observed for the soil-associated (soil cluster) strains. In contrast, a higher G+C content was found in the bacteria belonging to the soil cluster, as compared to other clusters (Table 3, Appendix A). For the soil clade, eight heat-shock-related genes (four ipbA genes—*HR0375, HR1773, HR2572*, and *HR2592*—and two Hsp20-encoding genes—*HR2563* and *HR2573*) and 14 DNA-repair-related genes were found in the genome of *M. thermotolerans* HR1, and similar genes were detected in the genome of *M. subterranea* Fail4, indicating their thermo-tolerance potential. Additionally, a UV damage repair endonuclease (UvdE) and DNA mismatch repair proteins (MutS, MutS2, and Mutl) were found in the genome of *Microvirga* sp. 17 mud 1–3. Meanwhile, DNA recombination repair pathways were also discovered in *M. thermotolerans* HR1. Furthermore, five strains, i.e., *M. lotononidis* WSM3557, *M. ossetica* V5/3M, *M. vignae* BR3299, *Microvirga* sp. KLBC81, and *M. guangxiensis*, formed a cluster named the root nodule clade, which had nitrogen-fixing and/or -forming nodules in common. The exception to this division was the soil-isolated strain *M. guangxiensis*, which functionally clustered with root nodule isolates (Figure 4b). Generally, *nif* genes and a nitrogen fixation regulator all existed in the genome of these strains. Moreover, the genes involved in nodulation formation (e.g., the *nod* genes) were also found in their genomes (Appendix A). Interestingly, the *nod* genes required for nodulation was absent in *M. ossetica* V5/3M and *M. guangxiensis* CGMCC1.766. In addition, three independent clusters were separated according to isolation source (air, human stool, or rapeseed endophyte, respectively) (Figure 4b). A series of genes coding for phosphorus and potassium-solubilization protein were found in *M. brassicacearum*, indicating its role in plant growth promotion. In the genome of *Microvirga massiliensis* JC119, which was isolated from human stool, key genes involved in heme synthesis, transport, and secretion during infection implied the potential for *Microvirga massiliensis* JC119 to result in disease [29,30].

Moreover, the linear comparison based on the three complete genomes of *Microvirga* indicated that *M. thermotolerans* HR1 was more linear with the *Microvirga* sp. 17 mud 1–3 derived from soil than the root nodule isolate *M. ossetica* V5/3M (Appendix A).

### 3.5. Description of Microvirga thermotolerans sp. nov.

*Microvirga thermotolerans* (ther.mo.to′le.rans. Gr. fem. n. therme heat; N.L. pres. part. tolerans tolerating; N.L. part. adj. thermotolerans heat-tolerating) cells are Gram-stain-negative, aerobic, single polar flagellum, and rod-shaped (1.2–0.7 × 2.8–0.9 µm). Growth occurs on R2A agar, but not on LB and TGY. Colonies on R2A agar are milk white, semi-transparent, smooth, drop-shaped, and smaller than 0.5 mm in diameter after 3 days at 40 °C. The strain HR1 produces milk-white colonies, and growth occurs within a range of 15–50 °C, pH 6.0–8.0, and 0–0.5% (*w/v*) NaCl. The catalase is positive, and the oxidase is negative. The catalase is positive for alkaline phosphatase, esterase (C4), and weakly positive for esterase lipase (C8), negative for lipase (C14). The major respiratory quinone is Q-10, and the major fatty acids are summed feature 8 (C_18_:_1_ω7c and/or C_18_:_1_ω6c (56.76%) and C_19_:_0_ cyclo ω8c (16.8%)). The polar lipids contain diphosphatidylglycerol (DPG), phosphatidylethanolamine (PE), phosphatidylglycerol (PG), phosphatidylcholine (PC), phospholipids (PL), and aminolipid (AL). The genomic DNA G+C content of the type strain is 67.71%. The GenBank accession numbers for the 16S rRNA gene sequence and whole genome sequence of the type strain HR1 are MN524586 and CP045423, respectively. The type strain, HR1^T^ (=GDMCC 1.1658^T^ = KACC 21422^T^), was isolated from rice rhizospheric soil from Wuchang, a traditional rice-planting area in the northeast of China.

## 4. Discussion

*Microvirga* spp. are Gram-negative α-proteobacteria that are found in various environments. Members of the *Microvirga* genus show a diverse spectrum of metabolic activities, which is indicative of their adaptation to various niches such as soil, air, and human hosts [31,32]. *Microvirga* spp. have also been found to be nitrogen-fixing rhizobia and plant growth-promoting endophytic bacteria [10,12,14]. In this study, a novel thermo-tolerant strain named *M. thermotolerans* HR1 was isolated from rice paddy. The 16S rRNA sequence analysis revealed maximum identity with *Microvirga* sp. 17 mud 1–3 (97.87%). Additionally, the strain HR1 contained highly similar components of quinone and main fatty acids with respect to other members of the *Microvirga* genus [4]. According to the principles defined by Jongsik and Aharon [33], bacterial 16S rRNA gene sequence with a similarity under 98.7% with respect to its closest related species represented a novel species. *M. thermotolerans* HR1 differed from the closest species in many features. For example, nitrate reduction was negative for *M. thermotolerans* HR1 but positive for its relatives, *Microvirga flavescens* c27j1. In addition, *M. thermotolerans* HR1 presented milk white colonies on an R2A agar medium, while two close species *M. flavescens* and *M. aerilata* 5420S-16 presented light yellow and pink colonies, respectively [3,4]. Moreover, the predominant polar lipids in *M. thermotolerans* HR1 also showed differences from other relatives. The digital DNA–DNA hybridization values of strain HR1 and *Microvirga* sp. 17 mud 1–3 and of that and *M. flavescens* c27j1 were 38.2 and 19.9%, respectively. According to the definition of a novel bacterial species (ANI was lower than 96–98%, and dDDH was lower than 70%) and the principles described by Varghese [34] and Stackebrandt and Goebel [35], combined with the phenotypic and biochemistry data, strain HR1 should be classified as representative of a novel species of the *Microvirga* genus. Among all members of the *Microvirga* genus, the highest thermo-resistance was only found in strains *M. thermotolerans* HR1 and *M. subterranean* FaiI4 [2], which survived temperatures of 40 °C. When the temperature reached 45–50 °C, *M. thermotolerans* HR1 showed a higher tolerance than *M. subterranean* FaiI4. Considering the fact that this strain was isolated at 48 °C and that it expressed biological traits at temperatures ranging from 15 to 50 °C, it is more appropriate to call it a thermo-tolerant bacterium rather than thermophilic (heat-loving) [36]. Indeed, key genes of strain HR1, which were responsible for heat shock response and DNA recombination repair, were annotated during genome analysis. Compared with *M. subterranea* FaiI4, both of their genomes contained the Hsp family genes. In addition, strain HR1 contained more stress-resistance-related genes than the strain *M. subterranea* FaiI4. It is well known that Hsps can be induced by heat shock, and Hsp20 is known to be a small heat shock protein in the radioresistant bacterium *Deinococcus radiodurans* [37]. In addition, DNA recombination repair pathways were discovered in *M. thermotolerans* HR1. The MutL-MutS pathway, for example, which contains DNA mismatch repair proteins, performs a central role in bacteria such as *Microvirga* sp. 17 mud 1–3 [15], *Deinococcus radiodurans* [15], *E. coli* [38], and *Salmonella serotypes* [39]. Another example of a DNA repair system is the RecF pathway. For *E. coli*, eight proteins in the RecF pathway, namely RecA, RecN, RecF, RecO, RecR, RecQ, RecJ, and SSB [40], which are responsible for the recombinational repair of DNA damage, are also included in the genome of *M. thermotolerans* HR1. Thus, it was speculated that the thermo-tolerance capacity of *M. thermotolerans* HR1 was mainly attributed to the existence of heat shock response and DNA-repair-function-related genes.

Genome phylogenetic analysis based on 11 strains of the *Microvirga* genus generated five clades depending on isolation source, including human stool, soil, air, root nodule, and rapeseed endophytes, respectively. *M. thermotolerans* HR1 fell into the soil cluster which shares niche-specific functions (e.g., thermo-tolerance and radiation resistance) with *M. subterranea* FaiI4 and *Microvirga* sp. 17 mud 1–3. This is consistent with the phenotypic observation above. Similarly, the largest clade, the root nodule, was formed of five species with the abilities of nitrogen fixation and/or nodulation formation. Although *Microvirga* is not a close relative of the *Rhizobium* genus [10], it is interesting that a small fraction (four out of the 11) of the *Microvirga* spp. was shown to be symbiotic nitrogen-fixing bacteria. This suggests that such gene content in five species of *Microvirga* might be obtained from rhizobia. Previous work has indicated that rhizobia may sense rhizosphere environments and transfer gene content to other genera [41]. Moreover, *M. brassicacearum* CDVBN77 has been described as a plant endosymbiont capable of promoting plant growth by providing nutrients to hosts [14]. Given that members of this clade contain nodulation-forming and nitrogen-fixing genes, *Microvirga* species may play an important role in plant–microbe interaction. It is noted that, a trend toward a higher G+C content has been observed in the soil clade, possibly as a result of a more varied environment which means higher chance for horizontal gene transfer. Indeed, based on genetic analysis, the core genome of the *Microvirga* genus was conserved, and the pan genome was open, leading to a high possibility that foreign genes integrated into the genome by horizontal gene transfer over years of evolution. Therefore, the distribution pattern of the *Microvirga* genus reflects a high correlation between functional properties and their respective environments.

In this study, we report a novel species of the *Microvirga* genus, and comparative genomic analysis revealed the niche adaptation of *Microvirga* species. Genome phylogenetic analysis generated five clades that suggested a niche-specific adaption in the *Microvirga* genus. The results have the potential to provide information that facilitates future studies relating to the cloning and functional analysis of genes in *Microvirga* species.

## Figures and Tables

**Figure 1 microorganisms-08-00101-f001:**
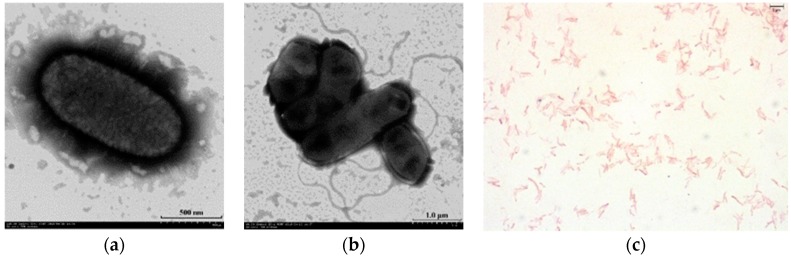
Transmission electron micrograph showing the morphology of strain HR1 and Gram-stain of HR1. Bar: 500 nm for (**a**), 1.0 μm for (**b**), 5 μm for (**c**).

**Figure 2 microorganisms-08-00101-f002:**
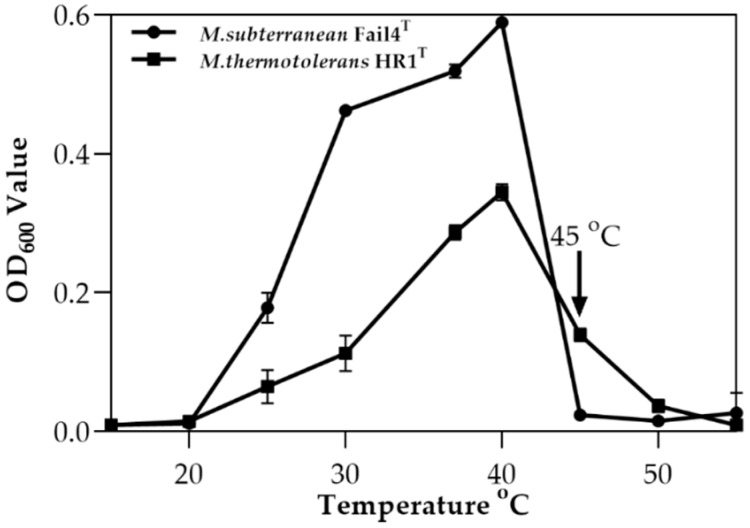
*M. subterranean* FaiI4 and *M. thermotolerans* HR1 strains culture for 15 h under different temperatures at a pH fixed at 7.0 ± 0.2. The values are means ± standard deviations (*n* = 4).

**Figure 3 microorganisms-08-00101-f003:**
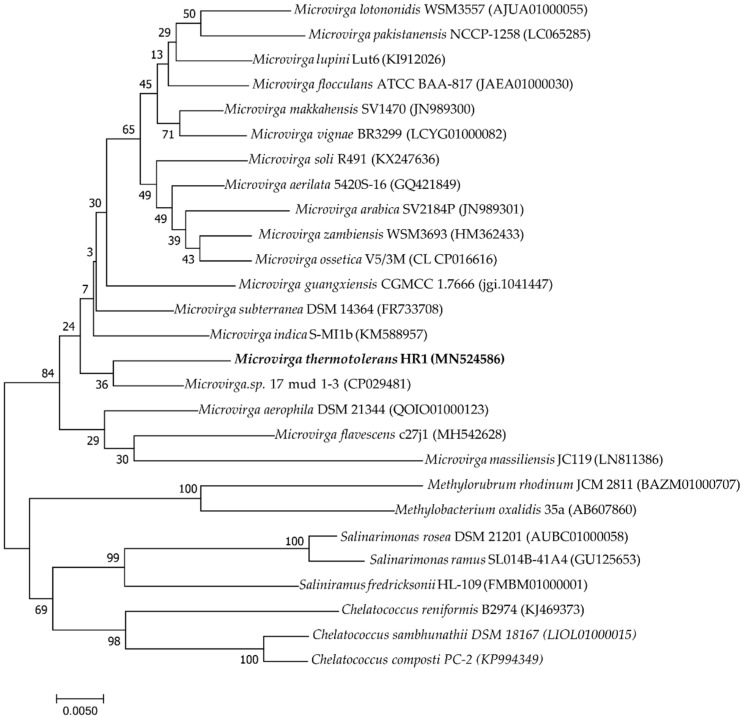
Phylogenetic tree based on 16S rRNA gene sequences reconstructed by the neighbor-joining method. This tree shows the phylogenetic relationship between strain HR1 and closely related species. Bootstrap percentages are based on 1000 replications. The GenBank accession numbers for 16S rRNA gene sequences are shown in parentheses. Bar: 0.005 substitutions per nucleotide position.

**Figure 4 microorganisms-08-00101-f004:**
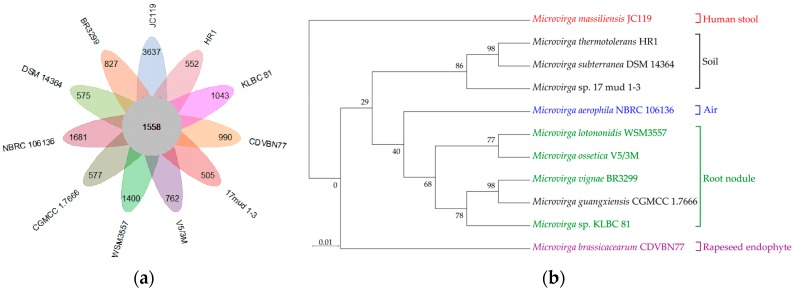
The Pangenome analysis and the genome phylogenetic tree of strains belonging to the *Microvirga* genus. (**a**) Petal diagram of the pangenome. Each strain is represented by a colored oval. The center is the number of orthologous coding sequences shared by all strains (i.e., the core genome). Numbers in nonoverlapping portions of each oval show the numbers of CDSs unique to each strain. The total numbers of protein-coding genes within each genome are listed in Table 3. (**b**) Groups divided into five groups based on the phylogenetic tree, which is based on 1558 core orthologous proteins of the *Microvirga* genus. Bootstrap values are expressed as percentages of 1000 replications. Bar: 0.01 substitutions per amino acid.

**Table 1 microorganisms-08-00101-t001:** Differential physiological characteristics between strain HR1 and closely related type strains and type species of the genus *Microvirga* (Strains: 1. *M. thermotolerans* HR1 (data from this study); 2. *M. flavescens* c27j1 (data from this study); 3. *M. indica* S-MI1b (data from this study); 4. *M. subterranean* FaiI4 (data from this study); +: positive; w: weakly positive; −: negative; Nitrate reduction, hydrolysis and assimilation test by using API 20NE, Enzyme activity test by using API ZYM, carbon source utilization test by using Biolog.

Characteristic	1	2	3	4
Isolation source	Rice paddy	Forest soil	Soil	Thermal aquifer
Cell dimensions (µm)	0.5–0.9 × 1.2–2.8	0.4–0.7 × 1.8–4.2	0.6–0.65 × 0.9–1.2	1.06 × 1.5–4.0
Motility	+	+	+	+
Conditions for growth:				
Temperature range (°C)	15–50	15–37	25–45	25–45
Optimum temperature (°C)	40	30	40	41
NaCl tolerance (%, *w/v*)	0–0.5	0–1	0–7	<1.0
pH range	6–8	6–10	6–12	6–9
Nitrate reduction	−	−	+	+
Hydrolysis of gelatin	−	−	−	+
Assimilation:				
Arabinose	−	−	w	+
Citric acid	−	−	+	−
Phenylacetic acid	−	+	−	−
Enzyme activity:				
Alkaline phosphatase	+	+	+	+
Acid phosphatase	w	+	+	+
Esterase(C4)	+	+	+	+
Esterase lipase (C8)	w	+	+	+
Lipid enzyme (C14)	−	−	−	−
Carbon source utilization:				
α-d-Glucose	−	w	w	−
D-Fucose	−	w	w	−
Fusidic Acid	−	w	w	w
Myo-Inositol	−	w	w	−
Minocycline	w	−	−	−
Tetrazolium Violet	+	w	w	w
α-Ketoglutaric Acid	−	w	w	w
Tween 40	−	w	+	w
DNA G+C content (%)	67.7	62.2	67.2	65.1

**Table 2 microorganisms-08-00101-t002:** Fatty acid composition of strain HR1 and closely related species and type species of the genus *Microvirga.* (Strains: 1. *M. thermotolerans* HR1 (data from this study); 2. *M. flavescens* c27j1 (data from this study); 3. *M. indica* S-MI1b (data from this study); 4. *M. subterranean* FaiI4 (data from this study); * Summed features are groups of two or three fatty acids, which cannot be separated using the MIDI system. Summed feature 2 contains C_12:0_ aldehyde and/or unknown ECL 10.9525; summed feature 3 contains C_16:1_ ω7c/C_16:1_ ω6c and/or C_16:1_ ω6c/C_16:1_ ω7c; summed feature 7 contains C_19:1_ ω7c/C_19:1_ ω6c and/or C_19:1_ ω6c/ω7c/19cy; summed feature 8 contains C_18:1_ ω7c and/or C_18 1_ ω6c.

Fatty Acid	1	2	4	5
C_16:0_	8.17	6.12	13.87	7.09
C_17:0_	2.78	-	0.46	12.14
C_18:0_	4.57	2.81	4.52	3.67
C_17:0_ cyclo	-	2.65	-	3.46
C_19:0_ cyclo ω8c	16.8	43.12	4.52	20.46
C_20:2_ ω6,9c	0.74	2.12	-	0.8
11-Methyl C_18:1_ω7c	0.85	2.65	-	1.44
C_14:0_ 2-OH	2.03	-	-	-
C_18:0_ 3-OH	1.48	2.25	1.77	1.26
* Summed feature 2	2.16	5.57	2.96	2.2
* Summed feature 3	1.36	2.47	3.68	0.84
* Summed feature 7	2.28	0.55	-	-
* Summed feature 8	56.76	28.7	68.49	43.72

**Table 3 microorganisms-08-00101-t003:** General features of bacterial genomes used in this study.

Strain Name	Accession No.	Level	Contigs	N50	L50	Size (Mb)	GC%	No. of Genes	No. of Proteins	Isolation
*Microvirga massiliensis* JC119	LN811386	Scaffold	365	56347	49	9.1	62.7	8575	8045	Human stool
*Microvirga vignae* BR3299	LCYG01000000	Scaffold	165	99306	19	6.47	61.1	6304	5674	Nodule
*Microvirga subterranea* DSM 14364	QQBB01000000	Scaffold	36	519753	4	5.15	65.1	4975	4902	Thermal aquifer
*Microvirga aerophila* NBRC 106136	BJYU00000000	Contig	628	43211	42	7.36	61.6	7220	7162	Air
*Microvirga guangxiensis* CGMCC 1.7666	FMVJ01000000	Scaffold	34	407384	5	4.72	61.4	4572	4499	Paddy soil
*Microvirga lotononidis* WSM3557	AJUA01000000	Scaffold	104	261220	10	7.08	63.1	6915	6524	Nodule
*Microvirga brassicacearum* CDVBN77	VCMV01000000	Scaffold	78	150733	10	5.22	62.3	4953	4755	Rapeseed
*Microvirga ossetica* V5/3M	CP016616	Complete	-	-	-	5.84	63.5	5543	5265	Nodule
*Microvirga* sp. 17 mud 1-3	CP029481	Complete	-	-	-	4.4	64	4143	4001	Soil
*Microvirga* sp. KLBC 81	QDAH01000000	Scaffold	357	70985	26	6.86	61.3	6588	6085	Nodule
*Microvirga thermotolerans* HR1	CP045423	Complete	-	-	-	3.8	67.71	3875	3818	Soil

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
