# Peer review of "Isolation and Identification of Microvirga thermotolerans HR1, a Novel Thermo-Tolerant Bacterium, and Comparative Genomics among Microvirga Species"

_microorganisms, 2020, doi:10.3390/microorganisms8010101_

Round 1
Reviewer 1 Report
This manuscript described a novel strain as a thermophilic bacterium isolated from rhizosphere soil as a non-extreme environment. Why did you isolate your strain under high temperature from non-thermo sample? In addition, compare to other studies to related novel taxonomy, this study seems to be preliminary and poor interpretation.
In the MS, there is many typos. Before submission, carefully read and check that.
L33-35, re-cite references. E.g Kanso and Patel [2], in addition, you should describe the genus emended information.
L36-39, there are 16 validated strains. You have to describe all that.
L42-43, many features are common in the genus Microvirga. If yes, which are significantly differences?
L43-46, extremely poor description. Rewrite.
L49, t? it might be typo.
L33-60, have to upgrade your introduction including your aims. The reason why you tried to isolate and describe the strain from non-extreme environment are still unclear.
L64, provide GPS
L63-70, have to describe the processing for isolation and purification. In addition, change to pH 7.0.
L70, change to 16S rRNA gene sequencing. but, please rewrite.
L71, change to analysis
L73, remove nearly complete
L75, check reference style
L72-83, how did you determine the full-length gene sequences for 16S rRNA and some housekeeping?
L85-86, your strain had polymorphisms under different cultivation? If not, rewrite.
L91, again reference style, please check that, carefully on your whole MS.
L92, with strain HR1?, rewrite.
L94, what’s interval?
L95, 0 to 0.5, interval 0.5? also, did you use buffers? Please provide the used buffer.
L96, 40C is your optimal temperature?
L96-97, during anaerobic culture, what kind of electron donors and acceptors were used?
L100, you have to use 20NE not 20E, re-do and re-description. Also, you have to analyze biolog plate or 50CH (API).
L104-105, what were your reference strains? And how did you determine the reference strains?
L110, remove analysis. And rewrite your subheading
L113-114, rewrite.
L115, HiSeqX10?
L117-118, rewrite.
L135-126, no, you have to submit and get the freely access number on public database such as GenBank.
L127, comparative genome analysis with who?
L134-5, preliminary typo.
L133-140, these strains can do sporulation. If yes, you should describe and provide the result. If not, remove this experiment. You have to test more widely ranged temperatures.
L142, omit organism and rewrite such as isolation and phylogenetic analysis for 16S rRNA and housekeeping genes.
L143, duplicate with M&M, rewrite.
L145, why did you use rouf’s fluide medium? Your strain could not grow under R2A sole medium?
L145-155, poor interpretations. Rewrite.
L156, poor English for subheading
L160, how did you determine gram type and motility?
L167, change to N-acetyl…..
L163-170, carefully check the write for each substrate such as upper and/or lower cases
L175-176, provide the difference proportions for lipid composition.
L177, remove comma before (56.76%)
L177-180, check L45.
L183, remove mol
L183, insert genes after rRNAs.
L185, 9 Mbp genome size? Extremely wide range genome size. Please carefully check and provide the genomic characters as table. It will be more helpful to readers’ understanding.
L184-188, which one corrected present? Figure or fig.
L188-191, in your result, there is no similarity (value) results for 16S rRNA and housekeeping genes, even, who is or are the closest strain(s)? then, we can analyze ANI and dDDH. w/o similarity analysis, how can we specify who can be my closest strains? My recommendation you have to do and provide the results for wet-DDH with your all closest strains.
L191-193, no. there is no scientific reasons.
L194-239, combined into one category. And from the analysis, do you identify any scientific clues? Sorry, I can pick up any scientific interesting. Seems to be preliminary data. In this part, you just provided simple locus tags of your genome. It’s nothing.
L240, it’s curious. Your strain can grow up to 50C. but, in L245, just 30% cell can survive?
L247, sorry I am not a nomenclature, but your strain is not thermotolerans. So, please reconsider it.
L253, remove the sentence. It’s your part for description for only your stain.
L257-264, carefully check the typo for substrates.
L268, remove mol
L273-275, remove. Using single strain, it’s difficult to amendments.
Fig.1 poor phylogenetic tree. Remove less 50%. You already mentioned in L282-283.
L283, 0.01 or 0.005?
Fig.2. where is your bar and flagella or flagellum?
Table 1. remove identical result and improve (insert and present) in table legend.
Table 2. provide summed features. And which means asterisk?
Fig 3. Poor interpretation. Improve figure legend. And the axis of (b) indicates for what?
Fig. 4, what are differences between unique and accessory gene contents?
Fig. 5. Please consider remove them.
L321, please combine into results and discussion. In discussion there is many duplicates.
Fig s1 and s3 combined into single figure. In addition, you should provide phylogenomics tree for the genus Microvirga.
Fig. S4. Provide separated lipid results such as total lipids, phospholipid, aminolipid, and glycolipid.
Table s1, provide genome completeness using checkm.
Tlable s2, poor presentation. Could not follow your meaning.
Fig. s5 consider to remove.
Author Response
Microorganisms 30-Dec-2019
Revision of microorganisms-666193: “Isolation, Identification of Microvirga thermotolerans HR1, a novel thermophilic bacterium and Comparative Genomics of the genus Microvirga” by Jiang Li, Ruyu Gao, Yun Chen, Dong Xue, Jiahui Han, Jin Wang, Qilin Dai, Min Lin, Xiubin Ke *, Wei Zhang *.
Dear Editor,
As request we have made a major revision on our manuscript according to the suggestions and comments by referees. We are very grateful for the constructive reviews. Besides, an intensive English-language improved has been done in the agency recommended by MDPI. The revision includes 26 text pages, 4 figures and 3 Tables, as well as a supplementary file. Below, we add a rebuttal letter, which addresses point by point the comments and suggestions by the referees.
Sincerely yours,
Prof. Dr. Wei Zhang
Biotechnology Research Institute, Chinese Academy of Agricultural Sciences, Beijing 100081, China
Tel +86-10-82106106
E-mail: zhangwei01@caas.cn
Response to the comments by referees
Comments of Reviewer 1:
This manuscript described a novel strain as a thermophilic bacterium isolated from rhizosphere soil as a non-extreme environment. Why did you isolate your strain under high temperature from non-thermo sample? In addition, compare to other studies to related novel taxonomy, this study seems to be preliminary and poor interpretation.
Reply: We agreed with your concerns. The first aim of our work was to isolate and identified bacteria from an over 100-year history paddy soil in the northeast of China. Actually, we got many isolates by incubating the soil under different conditions like temperature and nutrient situation. Microvirga thermotolerans HR1 was one of such isolates which could grow at temperature ranging from 15 – 50 ºC with a growth optimum at 40 ºC. Although due to the fact that M. thermotolerans HR1 showed higher tolerance than strain M. subterranean FaiI4 (Kanso and Patrel, Microvirga subterranea gen. nov., sp. nov., a moderate thermophile from a deep subsurface Australian thermal aquifer, IJSEM, 2003), it was more appropriate to call it a thermo-tolerant bacterium rather than thermophilic (heat-loving). So we changed descriptive title “thermophilic bacterium” to “thermo-tolerant bacterium” through whole manuscript. Then, studies of culturable bacteria belonging to the Microvirga genus typically focus on isolation, identification, and phenotypical characterization, while little attention has been paid to the genetic make-up of the Microvirga genus and to comparing their phenotypes. The present study describes the novel thermo-tolerant bacterium Microvirga thermotolerans HR1. Furthermore, the whole genome sequencing of the Microvirga thermotolerans HR1 strain and comparative genomic analysis of 11 strains of Microvirga are provided to explain the specific niche adaptation of various Microvirga strains.
In addition, the aim of this study had been remarked in the section “Introduction”, and more detail had been added in the section “M&M”. Furthermore, we made a major revision on the manuscript, including totally re-analyses the data especially for the comparative genomics part, re-writing the “Introduction” and “Discussion”, and organizing the structure of the figures and tables. All modifications had been highlighted.
In the MS, there is many typos. Before submission, carefully read and check that.
Reply: corrected.
L33-35, re-cite references. E.g Kanso and Patel [2], in addition, you should describe the genus emended information.
Reply: added. See L36-L42.
L36-39, there are 16 validated strains. You have to describe all that.
Reply: added.
L42-43, many features are common in the genus Microvirga. If yes, which are significantly differences?
Reply: added. The significant difference in the genus Microvirga was found in their physical characteristics shown in Table 1 and 2, and biological functions like symbiotic nitrogen-fixation and stress-resistance which showed high correlation with their respective environments.
L43-46, extremely poor description. Rewrite.
Reply: it has been rewritten.
L49, t? it might be typo.
Reply: corrected.
L33-60, have to upgrade your introduction including your aims. The reason why you tried to isolate and describe the strain from non-extreme environment are still unclear.
Reply: we have rewritten it in the “Introduction”.
L64, provide GPS
Reply: added.
L63-70, have to describe the processing for isolation and purification. In addition, change to pH 7.0.
Reply: added and corrected.
L70, change to 16S rRNA gene sequencing. but, please rewrite.
Reply: corrected.
L71, change to analysis
Reply: corrected.
L73, remove nearly complete
Reply: corrected.
L75, check reference style
Reply: corrected.
L72-83, how did you determine the full-length gene sequences for 16S rRNA and some housekeeping?
Reply: added more information.
L85-86, your strain had polymorphisms under different cultivation? If not, rewrite.
Reply: corrected.
L91, again reference style, please check that, carefully on your whole MS.
Reply: corrected.
L92, with strain HR1?, rewrite.
Reply: corrected.
L94, what’s interval?
Reply: corrected. The interval was described at L112
L95, 0 to 0.5, interval 0.5? also, did you use buffers? Please provide the used buffer.
Reply: corrected the interval, and we just added NaCl to Rouf’s liquid medium, no any other buffer was used.
L96, 40C is your optimal temperature?
Reply: yes. The growth of HR1 strain and the control strain were determined when incubated under different temperature conditions (15, 20, 25, 30, 37, 40, 45, and 50 °C).
L96-97, during anaerobic culture, what kind of electron donors and acceptors were used?
Reply: we evaluated the anaerobic growth capacity of strain HR1 by incubating it in MGC AnaeroPouch-Anaero according to the manufacturers’ instruction. The result showed the strain HR1 could not grow under anaerobic condition.
L100, you have to use 20NE not 20E, re-do and re-description. Also, you have to analyze biolog plate or 50CH (API).
Reply: we added the experiment and corresponding result.
L104-105, what were your reference strains? And how did you determine the reference strains?
Reply: for physiological and biochemistry characteristics, M. flavescens c27j1, M. indica S-MI1b and M. subterranean FaiI4 were chosen as reference strains. The strain M. flavescens c27j1 and M. indica S-MI1b was selected because of the similarity of 16S rRNA gene (except for Microvirga sp. 17 mud 1-3, but this strain can’t get from any organization); The M. subterranean FaiI4 was selected because it was the type species of Microvirga genus. To test the effect of temperature on bacterial growth, M. subterranean FaiI4 which was considered as thermophilic was selected as a control. And for the comparative genomic analysis, all available genome sequences (total 11) of strains were used.
L110, remove analysis. And rewrite your subheading
Reply: rewritten.
L113-114, rewrite.
Reply: rewritten.
L115, HiSeqX10?
Reply: rewritten.
L117-118, rewrite.
Reply: rewritten.
L135-126, no, you have to submit and get the freely access number on public database such as GenBank.
Reply: added the GenBank accession number.
L127, comparative genome analysis with who?
Reply: 11 whole-genome comparisons between M. thermotolerans HR1 and the publicly available genome sequences of members of the genus Microvirga were used for the comparative genomic analysis. Has been rewritten.
L134-5, preliminary typo.
Reply: rewritten.
L133-140, these strains can do sporulation. If yes, you should describe and provide the result. If not, remove this experiment. You have to test more widely ranged temperatures.
Reply: rewritten.
L142, omit organism and rewrite such as isolation and phylogenetic analysis for 16S rRNA and housekeeping genes.
Reply: rewritten.
L143, duplicate with M&M, rewrite.
Reply: rewritten.
L145, why did you use rouf’s fluide medium? Your strain could not grow under R2A sole medium?
Reply: added more description of experiment design. Actually, the strain could not grow under R2A fluid medium, although it could grow on R2A agar medium. The growth test was performed on R2A agar (and liquid), LB agar (and liquid), TGY agar (and liquid), and Rouf’s liquid medium (L108-109).
L145-155, poor interpretations. Rewrite.
Reply: rewritten.
L156, poor English for subheading
Reply: rewritten.
L160, how did you determine gram type and motility?
Reply: A method for gram-stain reaction determination. Fixed smear, cover smear with crystal violet solution (1 min), wash the stain off gently, cover the smear with gram's iodine solution for 1 min, wash the stain off with water, decolorize (solution, mix equal volumes of 95% ethanol and acetone) and counterstain smear (solution, mix 2.5 g safranin O with 100 ml of 95% ethanol), examine. We observed the strain own flagellum by transmission electron microscope, but not a motility-test experiment. So we removed this word.
L167, change to N-acetyl…..
Reply: corrected
L163-170, carefully check the write for each substrate such as upper and/or lower cases
Reply: rewritten.
L175-176, provide the difference proportions for lipid composition.
Reply: Corrected the description.
L177, remove comma before (56.76%)
Reply: corrected.
L177-180, check L45.
Reply: corrected.
L183, remove mol
Reply: corrected.
L183, insert genes after rRNAs.
Reply: corrected.
L185, 9 Mbp genome size? Extremely wide range genome size. Please carefully check and provide the genomic characters as table. It will be more helpful to readers’ understanding.
Reply: yes, the genome size was reported by Aurélia Caputo et al. (Microvirga massiliensis sp. nov., the human commensal with the largest genome, MicrobiologyOpen, 2016). In addition, we provided all genomic characters in Table 3.
L184-188, which one corrected present? Figure or fig.
Reply: corrected.
L188-191, in your result, there is no similarity (value) results for 16S rRNA and housekeeping genes, even, who is or are the closest strain(s)? then, we can analyze ANI and dDDH. w/o similarity analysis, how can we specify who can be my closest strains? My recommendation you have to do and provide the results for wet-DDH with your all closest strains.
Reply: we added similarity (value) results for 16S rRNA and housekeeping genes and analyze ANI and dDDH values was calculated by ANI calculator (www.ezbiocloud.net/tools/ani) and Genome-to-Genome Distance Calculator (GGDC 2.1) (http://ggdc.dsmz.de/home.php), respectively. These two tools have been widely used in many papers, e.g., Zhang and Zhang (Microvirga flavescens sp. nov., a novel bacterium isolated from forest soil and emended description of the genus Microvirga , IJSEM, 2019), and Liu and Shi (Neorhizobium lilium sp. nov., an endophytic bacterium isolated from Lilium pumilum bulbs in Hebei province, Archives of Microbiology, 2019).
L191-193, no. there is no scientific reasons.
Reply: rewritten.
L194-239, combined into one category. And from the analysis, do you identify any scientific clues? Sorry, I can pick up any scientific interesting. Seems to be preliminary data. In this part, you just provided simple locus tags of your genome. It’s nothing.
Reply: we have re-analyzed the genome data and made a major modification based on that.
L240, it’s curious. Your strain can grow up to 50C. but, in L245, just 30% cell can survive?
Reply: we have updated the result in Fig. 2.
L247, sorry I am not a nomenclature, but your strain is not thermotolerans. So, please reconsider it.
Reply: We agreed. We changed descriptive title “thermophilic bacterium” to “thermo-tolerant bacterium” through whole manuscript. Considering the fact that this strain was isolated at 48 °C and that it expressed biological traits at temperatures ranging from 15 to 50 °C, it is more appropriate to call it a thermo-tolerant bacterium rather than thermophilic (heat-loving).
L253, remove the sentence. It’s your part for description for only your stain.
Reply: corrected.
L257-264, carefully check the typo for substrates.
Reply: corrected.
L268, remove mol
Reply: corrected.
L273-275, remove. Using single strain, it’s difficult to amendments.
Reply: corrected.
Fig.1 poor phylogenetic tree. Remove less 50%. You already mentioned in L282-283.
Reply: corrected.
L283, 0.01 or 0.005?
Reply: corrected.
Fig.2. where is your bar and flagella or flagellum?
Reply: added.
Table 1. remove identical result and improve (insert and present) in table legend.
Reply: corrected.
Table 2. provide summed features. And which means asterisk?
Reply: added.
Fig 3. Poor interpretation. Improve figure legend. And the axis of (b) indicates for what?
Reply: corrected, we remove this figure and added a better figure.
Fig. 4, what are differences between unique and accessory gene contents?
Reply: We analyzed those data, and find out they were common in the COG and KEGG distribution, and largest proportion in the metabolic classification, our current topic is less about this result, so we removed this part.
Fig. 5. Please consider remove them.
Reply: removed.
L321, please combine into results and discussion. In discussion there is many duplicates.
Reply: we have rewritten the discussion.
Fig s1 and s3 combined into single figure. In addition, you should provide phylogenomics tree for the genus Microvirga.
Reply: we have added phylogenomics tree based on their whole-genome. Please see Fig. 4.
Fig. S4. Provide separated lipid results such as total lipids, phospholipid, aminolipid, and glycolipid.
Reply: added.
Table s1, provide genome completeness using checkm.
Reply: we provide the N50 and L50 value generated by Quality Assessment Tool for Genome Assemblies (QUAST) software to evaluate the assemblies quality. This value was used in many article, e.g., Miguel Rodríguez et al. ( Paenibacillus lutrae sp. nov., A Chitinolytic Species Isolated from A River Otter in Castril Natural Park, Granada, Spain, Microorganisms,2019). And Singh and Kaur (Antimicrobial properties of the novel bacterial isolate Paenibacilllus sp. SMB1 from a halo-alkaline lake in india, Scientific report, 2019)
Tlable s2, poor presentation. Could not follow your meaning.
Reply: upgraded.
Fig. s5 consider to remove.
Reply: removed.

Reviewer 2 Report
Journal: Microorganisms
Title: Isolation, Identification of Microvirga thermotolerans HR1, a novel thermophilic bacterium and Comparative Genomics of the genus Microvirga
Manuscript ID: microorganisms-666193
In this manuscript, a novel thermophilic strain, named HR1T, was classified in terms of genetic and physiologic features and was assigned to the genus Microvirga. The phenotypic, chemotaxonomic, and phylogenetic results suggest that the new strain HR1T should be classified as representative of a new species of the genus Microvirga, with the proposed name M. thermotolerans sp. nov. Type strain HR1T. Features of the new isolate, with resistance to high temperatures, could be used in biotechnological contextes.
The manuscript is interesting both from the tassonomic and the biotechnological point of view.
Does there are some additional strains of the new species?
Revisions
Line 17: “thermophilic bacterium” change to “thermophilic bacteria”;
line 37: “Microvirga aerophile” change to “Microvirga aerophila”;
line 44: “Chemotaxonomy” change to “Chemotaxonomically”;
line 57 “may represented” please, correct it;
line 63: “were collection” change to “were collected”;
line 64: “fluid” change to “liquid” and repeat the same correction (fluid change to liquid) throughout the text of the manuscript;
line 65: “culture solution”, possibly it would be a ‘culture suspension’. Actually, I would suggest to change “culture solution” to “enrichment culture”, and repeat where necessary throughout the text of the manuscript, as teh term ‘enrichment culture’ is used when arranging microbial cultures, using an environmental sample as inoculum, to allow growth and isolation of microorganisms growing in these conditions;
line 93: “The optimal temperatures for growth was carried out by growth …” change to “The optimal temperatures for growth were investigated by growth …” or “Test for optimal temperatures for growth was carried out by growth …”;
line 103: “To investigate the chemotaxonomic” change to “To investigate the chemotaxonomy” or “To investigate the chemotaxonomic features”;
line 111: “After the cells was cultivated …” change to After the cells were cultivated …” or “After the cells have been grown …”;
line 132: “The resistant ability of strain …” change to “The resistance ability of strain …”;
line 133: “The resistant ability …” change to “The heat stress resistance ability …”;
line 134: “… and thermotolerans” change to “… and M. thermotolerans”;
line 145: “fluid medium” correct as suggester for correction at line 64 (see above);
lines 191-192: “… phylogenicm physiology and chemotaxonomy characteristics, …” change to “… phylogenetic, physiological and chemotonomic characteristics, …”;
line 257: “… but oxidase …” change to “… and oxidase …”;
line 273: “Zhang et al.” when citing an Author in the text it is better to report “Zhang and coll.”, control throughout the text of the manuscript;
Figures: the symbols in figures 3a and 3b; and 4b; are too small and it is difficult to comprehend;
line 314: “microvirga” change to “Microvirga”;
lines 338 and 339: “Vargese N.J. [30] and Stackebrandt E. [31]” change to “Vargese and coll. [30] and Stackebrandt and Goebel [31]”, correct throughout the manuscript;
line 345: “was conserve” change to “was conserved”;
line 356: “Based on all evidences above …” change to “Based on all evidences reported above …”;
line 362: “Microvirga core …” change to “Microvirga thermotolerans HR1T core …”
line 365: “sp.” not in Italic style;
line 366: “serotypes” not in Italic style (and “serotypes” not in Italyc style in the Reference list, at reference number 36);
line 399: “TCL” change to “TLC”.
Author Response
Microorganisms 30-Dec-2019
Revision of microorganisms-666193: “Isolation, Identification of Microvirga thermotolerans HR1, a novel thermophilic bacterium and Comparative Genomics of the genus Microvirga” by Jiang Li, Ruyu Gao, Yun Chen, Dong Xue, Jiahui Han, Jin Wang, Qilin Dai, Min Lin, Xiubin Ke *, Wei Zhang *.
Dear Editor,
As request we have made a major revision on our manuscript according to the suggestions and comments by referees. We are very grateful for the constructive reviews. Besides, an intensive English-language improved has been done in the agency recommended by MDPI. The revision includes 26 text pages, 4 figures and 3 Tables, as well as a supplementary file. Below, we add a rebuttal letter, which addresses point by point the comments and suggestions by the referees.
Sincerely yours,
Prof. Dr. Wei Zhang
Biotechnology Research Institute, Chinese Academy of Agricultural Sciences, Beijing 100081, China
Tel +86-10-82106106
E-mail: zhangwei01@caas.cn
Comments of Reviewer 2:
In this manuscript, a novel thermophilic strain, named HR1T, was classified in terms of genetic and physiologic features and was assigned to the genus Microvirga. The phenotypic, chemotaxonomic, and phylogenetic results suggest that the new strain HR1T should be classified as representative of a new species of the genus Microvirga, with the proposed name M. thermotolerans sp. nov. Type strain HR1T. Features of the new isolate, with resistance to high temperatures, could be used in biotechnological contextes.
The manuscript is interesting both from the tassonomic and the biotechnological point of view.
Does there are some additional strains of the new species?
Reply: No. Many other strains have been isolated during our study. However, they were distributed in different generas, such as Acinetobacter, Bacillus and Paenibacillus.
Revisions
Line 17: “thermophilic bacterium” change to “thermophilic bacteria”;
Reply: corrected.
line 37: “Microvirga aerophile” change to “Microvirga aerophila”;
Reply: corrected.
line 44: “Chemotaxonomy” change to “Chemotaxonomically”;
Reply: corrected.
line 57 “may represented” please, correct it;
Reply: corrected.
line 63: “were collection” change to “were collected”;
Reply: corrected.
line 64: “fluid” change to “liquid” and repeat the same correction (fluid change to liquid) throughout the text of the manuscript;
Reply: corrected.
line 65: “culture solution”, possibly it would be a ‘culture suspension’. Actually, I would suggest to change “culture solution” to “enrichment culture”, and repeat where necessary throughout the text of the manuscript, as teh term ‘enrichment culture’ is used when arranging microbial cultures, using an environmental sample as inoculum, to allow growth and isolation of microorganisms growing in these conditions;
Reply: corrected. We use “enrichment culture” replace “culture solution” through whole manuscript.
line 93: “The optimal temperatures for growth was carried out by growth …” change to “The optimal temperatures for growth were investigated by growth …” or “Test for optimal temperatures for growth was carried out by growth …”;
Reply: corrected.
line 103: “To investigate the chemotaxonomic” change to “To investigate the chemotaxonomy” or “To investigate the chemotaxonomic features”;
Reply: corrected.
line 111: “After the cells was cultivated …” change to After the cells were cultivated …” or “After the cells have been grown …”;
Reply: corrected.
line 132: “The resistant ability of strain …” change to “The resistance ability of strain …”;
Reply: corrected.
line 133: “The resistant ability …” change to “The heat stress resistance ability …”;
Reply: corrected.
line 134: “… and thermotolerans” change to “… and M. thermotolerans”;
Reply: corrected.
line 145: “fluid medium” correct as suggester for correction at line 64 (see above);
Reply: corrected.
lines 191-192: “… phylogenicm physiology and chemotaxonomy characteristics, …” change to “… phylogenetic, physiological and chemotonomic characteristics, …”;
Reply: corrected.
line 257: “… but oxidase …” change to “… and oxidase …”;
Reply: corrected.
line 273: “Zhang et al.” when citing an Author in the text it is better to report “Zhang and coll.”, control throughout the text of the manuscript;
Reply: corrected.
Figures: the symbols in figures 3a and 3b; and 4b; are too small and it is difficult to comprehend;
Reply: We have adjusted the symbols.
line 314: “microvirga” change to “Microvirga”;
Reply: corrected.
lines 338 and 339: “Vargese N.J. [30] and Stackebrandt E. [31]” change to “Vargese and coll. [30] and Stackebrandt and Goebel [31]”, correct throughout the manuscript;
Reply: corrected.
line 345: “was conserve” change to “was conserved”;
Reply: corrected.
line 356: “Based on all evidences above …” change to “Based on all evidences reported above …”;
Reply: corrected.
line 362: “Microvirga core …” change to “Microvirga thermotolerans HR1T core …”
Reply: corrected.
line 365: “sp.” not in Italic style;
Reply: corrected.
line 366: “serotypes” not in Italic style (and “serotypes” not in Italyc style in the Reference list, at reference number 36);
Reply: corrected.
line 399: “TCL” change to “TLC”
Reply: corrected.

Reviewer 3 Report
Minor observations:
Line 63 - change collection with collected
Line 64 and other places - change fluid with liquid
Line 65 - "culture solution" it is inoculum? Why do you incubate 3 days and not 7 ? (i.e. Zhang et al., 2009, IJSEM journal)
Lines 94, 97 and 101 - appear cultivation for 7 days in contrast with cultivation for DNA isolation (see 2.1. and 2.5.)
Lines 143 - 144 - English check
Line 325 - add Jongsik and collaborators to reference list.
Author Response
Microorganisms 30-Dec-2019
Revision of microorganisms-666193: “Isolation, Identification of Microvirga thermotolerans HR1, a novel thermophilic bacterium and Comparative Genomics of the genus Microvirga” by Jiang Li, Ruyu Gao, Yun Chen, Dong Xue, Jiahui Han, Jin Wang, Qilin Dai, Min Lin, Xiubin Ke *, Wei Zhang *.
Dear Editor,
As request we have made a major revision on our manuscript according to the suggestions and comments by referees. We are very grateful for the constructive reviews. Besides, an intensive English-language improved has been done in the agency recommended by MDPI. The revision includes 26 text pages, 4 figures and 3 Tables, as well as a supplementary file. Below, we add a rebuttal letter, which addresses point by point the comments and suggestions by the referees.
Sincerely yours,
Prof. Dr. Wei Zhang
Biotechnology Research Institute, Chinese Academy of Agricultural Sciences, Beijing 100081, China
Tel +86-10-82106106
E-mail: zhangwei01@caas.cn
Comments of Reviewer 3:
Minor observations:
Line 63 - change collection with collected
Reply: corrected.
Line 64 and other places - change fluid with liquid
Reply: corrected.
Line 65 - "culture solution" it is inoculum? Why do you incubate 3 days and not 7 ? (i.e. Zhang et al., 2009, IJSEM journal)
Reply: We replaced “culture solution” by “enrichment culture” through whole manuscript. Because the soil sample incubated at 48 ºC for 3 days has been enough for the next step, this routine was repeated three times for get enough enrichment culture.
Lines 94, 97 and 101 - appear cultivation for 7 days in contrast with cultivation for DNA isolation (see 2.1. and 2.5.)
Reply: 7 days culture for strain HR1 is too long to extract good quality DNA by means of OD260/280 and OD230/260. The possible reason would be that secondary metabolites have been produced during incubation which is obvious under a microscope.
Lines 143 - 144 - English check
Reply: corrected.
Line 325 - add Jongsik and collaborators to reference list.
Reply: added.

Round 2
Reviewer 1 Report
most concerns have been addressed during revision. however, still, there is some errors.
e.g. L39, carefully check citation style.
L37-44, please remove "emended three times". rewrite the sentence. it's not interesting. according to increasing strain-isolation, the description of the genus might be changed based on the difference physiological characters.
L47-53, provide full name for your strains, on the first time.
L54-61, remove all, and please highlight specific and/or unique characters.
L69, remove "in ~ ecological niche". and replace by various environments.
L54-61 and l70-75, seems to be duplicate.
L84-88, still, i am unable to understand the culture temperatures. your strain is thermo-tolerant microbug. however, the reason why you used various temperature for cultivation is not still unclear.
L89, 16S rRNA gene...
L90, the optimal temperature of your strain was 40C. why did you use 48C?
L120, please present more scientific such as [0-0.5 % (w/v) NaCl, at interval of 0.1% (w/v)] or others...
L124-125, where was api 50CH or Biolog GEN III for various carbon/nitrogen substances?
in M&M, part 5 and 6 combine into single also, part 7 should be incoporated into part 3. in addition, why did you still carry on the result for temperature effect experiment? according to the culture temperature, your strain has been modified? if not, please consider remove.
L194-195, how did you determine the carbon utilizaiton even, tetrazolium ivolet? it's from biolog.
L213-215, typo. in italic
L281, move and locate the last section in your paper. following discussion. and "nov." not "Nov."
L284, how did you determine motility?
L297, please provide the certificates.
table1, still there is presenting identical results.
table3, please present complete or draft not scaffold or contig. and remove 50 and L50. provide the number of tRNA, rRNA or other genomic traits.
figure3. still presented poor scores such as 30, 3, 7, ... and others.
figure4. please remove clades. still poor bootstrrap values. in addition, root nodule might harbor endophyte.
L351, 3. discussion? change to 6, probably. in addition, this discussion part have to incorporate into result part. there is no discussion.
L352, gram-staining negative, Alphaproteobacteria - typo.
L353-355, wide spectrum of metabolic activity could not match to adaptation to their isolation. if yes, you have to discuss and pick up the any clues.
L361, citation style...
for supplementary figures, please remove low values for bootstrap...